# Valorization of Kiwi Peels: Fractionation, Bioactives Analyses and Hypotheses on Complete Peels Recycle

**DOI:** 10.3390/foods11040589

**Published:** 2022-02-18

**Authors:** Francesco Cairone, Stefania Garzoli, Luigi Menghini, Giovanna Simonetti, Maria Antonietta Casadei, Laura Di Muzio, Stefania Cesa

**Affiliations:** 1Department of Drug Chemistry and Technology, “La Sapienza” University of Rome, 00185 Rome, Italy; francesco.cairone@uniroma1.it (F.C.); stefania.garzoli@uniroma1.it (S.G.); mariaantonietta.casadei@uniroma1.it (M.A.C.); laura.dimuzio@uniroma1.it (L.D.M.); 2Department of Pharmacy, University “G. d’Annunzio”, Botanic Garden “Giardino dei Semplici”, 66100 Chieti, Italy; luigi.menghini@unich.it; 3Department of Environmental Biology, “La Sapienza” University of Rome, 00185 Rome, Italy; giovanna.simonetti@uniroma1.it

**Keywords:** kiwi peels, SPME-GC-MS analysis CIEL*a*b* analysis, HPLC-DAD analysis, DPPH assay, anti-*Candida* activity, circular economy

## Abstract

Kiwi fruit samples (*Actinidia deliciosa* Planch, cv. Hayward) represent a suitable and good source for fibers obtainment as well as for polyphenolic and carotenoid extraction. With this aim, in this study they were submitted to a double phase extraction to separate insoluble fibers by an organic phase containing lipophilic substances and an hydroalcoholic phase containing polyphenols and soluble fibers. Insoluble fibers could be separated by filtration and sent to be micronized and reused. Hydroalcoholic fractions were then furtherly fractionated by solid-phase extraction. Data coming from the color CIEL*a*b* and the HPLC-DAD analyses of the extracts were compared and correlate with those coming from the SPME-GC/MS analysis of either the finely shredded peels or of the extracts. The obtained extracts were also submitted to anti-radical activity evaluation and anti-*Candida* activity. Results show that all of the obtained residues are value added products. Hypotheses were also made about the nature and the possible recycle of the obtained purified solid residue.

## 1. Introduction

The term ‘kiwi’ indicates fruits produced by plants belonging to the Actinidiaceae family and specifically of the *Actinidia* Lindl genus, which are characterized by edible fruits, coming mainly from *Actinidia chinensis* var. hispida C.F. Liang (synonym *A. deliciosa* Chev., C.F. Liang, and A.R. Ferguson) *A. chinensis* and *A. arguta* (Siebold and Zucc.) Planch ex Miq., which represent the most commercialized. Some other less famous species, such as *A. kolomikta* (Maxim and Rups) Maxim and *Actinidia arguta* var. purpurea (Rehder) C.F. Liang and Q.Q. Chang (synonym *A. purpurea*, Rehder) also produce edible fruits [1].

More than fifty wild species are known [2], but only a few of these have been cultivated. The most diffuse cultivated plant of this genus is *Actinidia deliciosa*, whose production leaders are represented by China and Italy. Italy is also, along with New Zealand, one of the two biggest exporters of kiwis in the world. Kiwi composition boasts all of the best nutritional principles, such as proteins, lipids, carbohydrates and dietary fibers, vitamins, minerals polyphenols, and other antioxidants. Due to these properties and to its limited seasonal availability, other than as a fruit it is largely consumed as juice or puree, stored as frozen and lyophilized products, or finally used in fortified drinks and alcoholic beverages [3].

Valuable edible kiwi fruits come from the cultivar *A. deliciosa* “Hayward”. Fiorentino et al. [4] reported that fruits coming by eight different *Actinidia* genotypes were tested in relation to their polyphenols and vitamin C content, showing that the antioxidant capacity is largely influenced by species and cultivar. *A. deliciosa* “Hayward”, besides other cultivars, are also known for their positive health potential, show interesting properties due to the strong antioxidant, anti-inflammatory, and anti-diabetic capacity, combined with their antimicrobial, antiviral and antifungal activities. Most of these health properties are attributed to polyphenolic components such as phenolic acids (protocatechuic, gallic, caffeic, syringic, ferulic, salicylic, coumaric) and flavonoids (quercetin, procyanidins, chrisin and rutin), as well as a potent immunomodulatory effect ascribable to the very high vitamin C content [3].

Polyphenols and antioxidant molecules, are known to be present in the fleshly internal part of the fruits as well as in the peels, are considered a non-edible part of the fruit. Whole fruit *A. arguta,* also known as the mini kiwi, (and sometimes of *A. chinensis*, or golden kiwi) can be eaten. In this case, the smooth peels could also represent an edible part, as well as the peels of the most diffuse *A. deliciosa*, which typically represent a waste material [1]. Kiwi peels that are separated and discarded for fresh fruit consumption, as well as in industrial processing, represent an interesting source of relevant biomolecules that are only marginally investigated [4].

In this context, and according to the principles of the circular economy, great interest of the scientific community nowadays is devoted to the recycling and the valorization of agri-waste products to transform them into added value by-products. As reported by Aureli et al. [5], in a study on the food waste of an Italian family, “reducing food waste along the entire food supply chain is an important policy priority included in the United Nations Sustainable Development Goals for 2030”.

As with many others vegetal matrices, kiwi fruits and relative by-products contain interesting phytocompounds, whose recovery could represent a strategy to valorize the production and define innovative applications of economic relevance. They could also apply eco-friendly procedures in order to assure no risk for the consumers or to the environment.

The seeds can be conveniently reused as raw material for the extraction of a secondary product (such as the oil). It is noteworthy that the seeds are considered to be within the edible part of the fruit but during production of juice and jam they represent a relevant part of waste production [6,7,8].

Kiwi peels, which contain a high quantity of bioactive molecules such as polyphenols and antioxidants, represent one of the most important kiwi by-products to be recycled and valorized. In the last few years, kiwi peels extract has been successfully used to fabricate nanoparticles using a cost effective and eco-friendly green method of synthesis. Zinc oxide (ZnO), titanium dioxide (TiO_2_), and tin oxide (SnO_2_) nanoparticles (NPs) were prepared by exploiting the polyphenolic component of kiwi peels used as reducing and capping agent for NPs preparation and stabilization [9,10,11,12].

In this way there is the double advantage of recycling and valorizing kiwi peels, limiting the use of traditional toxic reducing and capping agent responsible of serious environmental pollution issues. The good antimicrobial activity and the anti-cancer activity sometimes exhibited by these nanoparticles, also makes them promising candidates for biomedical applications. In addition, polyphenol-rich kiwi peel extracts (PE) were also employed as reducing and capping agents for the synthesis of self-assembling silver@PE nanoparticles (Ag@PE NPs) incorporated into polymeric sodium alginate (SA)-based films. The presence of Ag@PE NPs successfully improved the antioxidant and antibacterial functions of SA-based films making them potential and promising candidates as multifunctional packaging films in food preservation field [13].

Different kiwi by-products for an optimized flowchart were recently revised by Sanz et al. [14]. A wide set of biological activities were reported, including the reduction of platelet aggregation, the normalization of triglycerides blood levels, protective effects against degenerative, cardiovascular, and cancer disease, efficacy in constipation as antiglycation and as nephroprotection, and their well-known antimicrobial activity. Great attention should be devoted to agricultural practices and processing due to their influence on the bioactive content of the starting materials, the health potential of the obtained products, and their impact on the environment. 

A limited number of published papers have explored the effects of the extraction procedures applied to the kiwi peels to the chemical quality and biological activities of extracts. Conventional extraction based on ethanol or acetone mixed with water were applied for the flavonoids extraction and acidic hydrolysis has been applied for the pectin’s extraction [15,16,17]. The same classes of compounds were also obtained by ultrasound and microwave assisted methods [18]. The selectively extracted fractions were then analyzed and compared to evaluate the optimal extraction method in terms of yield and biomolecules preservation. In a previous work, we tested with success a double phase extraction to goji berries. In fact, these are characterized, as kiwi fruits and peels, by a lipophilic and a hydrophilic fraction which is simultaneously represented. This approach allowed us to simplify the process, saving time, money, and processed wastes [19,20]. In the present work, the kiwi peels were chosen as starting material to study a process of complete recycle of this interesting and overproduced agri-waste. ‘Hayward’ kiwi fruits were selected for the present study coming directly from either an Italian plantation or from a commercial sample. Markets indicate that in Italy, which represents the first European producer of kiwis, about 400 thousand tons of kiwi fruits per year are produced and about the 10% of the production is represented by discarded peels. A flowchart for the kiwi peel valorization and reuse, based on the circular economy principles, is presented and evaluated in this study.

## 2. Materials and Methods

### 2.1. Standards and Reagents

Bidistilled water, acetone, ethanol, acetic acid, acetonitrile, *n*-hexane and methanol were purchased from Merck life Science s.r.l (Milan, Italy). Reference compounds for HPLC analysis, (+)-catechin, epicatechin, caffeic acid, benzoic acid, vanillic acid and 2,2-diphenyl-1-(2,4,6-trinitro-phenyl) hydrazine (DPPH) were purchased from Merck life Science s.r.l (Milan, Italy).

### 2.2. Sample Preparation 

Kiwi fruits cv. Hayward, were collected manually from an organic plantation in Campania Region, (Italy) consisting in adult plants productive since more than five years. Collection was carried out in July 2020, and fruit selection (Sample Series “**sel**”) was carried out on the basis of ripening stage defined by the farmers as optimal for the commercialization. A comparative sample of commercial fruits from conventional agriculture were purchased from local market. In the label, the producer defines the Hayward variety and the Italian origin (Sample Series “**com**”). Samples were stored at 4–6 °C for the time strictly necessary to complete the experimental procedures (about one week). Moisture content of peels was also evaluated (amounting to 78 ± 1%).

The fruit epicarps (peels) were manually separated from the internal fleshy pulp (consisting of mesocarps and endocarps) using a common potato peeler in order to obtain samples of uniform thickness. These were immediately homogenized by wet grinding, obtaining a paste with particles millimeters in size, which were submitted to extraction and forwarded to the experimental investigations. 

### 2.3. Solid Phase MicroExtraction (SPME) of Peels

The sampling by SPME technique was performed following Vitalini, et al. [21] with some modifications. Representative samples of peels (~2 g) were individually placed into a 20 mL glass vial with PTFE-coated silicone septum. 

For the extraction of volatiles compounds, a SPME device from Supelco (Bellefonte, PA, USA) with 1 cm fiber coated with 50/30 μm DVB/CAR/PDMS (divinylbenzene/carboxen/polydimethyl siloxane) was used. Before use, the fiber was conditioned at 270 °C for 30 min. Each sample was equilibrated for 30 min at 40 °C before sampling. Later, the fiber was exposed to the headspace of the samples for 30 min at 40 °C to collect and concentrate the volatiles compounds. Lastly, the SPME fiber was inserted in the GC injector maintained at 250 °C in split mode for the desorption of the captured components.

### 2.4. Double Phase Extraction 

For the extraction of the bioactive compounds from kiwi peels, a double phase extraction was performed according to our previous work [20]. About 25 g of kiwi peels were extracted with a 50 mL of *n*-hexane and 50 mL of a hydroalcoholic mixture (ethanol:water acidified with 5% of acetic acid, 70:30 *v*/*v*) for 3 h at room temperature under stirring. The two phases were separated and concentrated under reduced pressure at 40 °C with a rotary evaporator, weighed and stored at 4 °C until analyzed. The resulting extracts are identified as **HA_com_** and **HA_sel_** for hydroalcoholic fraction and **HE_com_** and **HE_sel_** for hexane fraction obtained from commercial sample or local cultivation, respectively.

### 2.5. Colorimetric Analysis

The extracts obtained from double phase extraction of peels, were submitted to colorimetric analysis at room temperature (20 °C ± 1), with a colorimeter X-Rite MetaVue^TM^, equipped with a full-spectrum LED illuminant and an observer angle of 45°/0° imaging spectrophotometer. The cylindrical coordinates were calculated according to a previous work [22]. pH of hydroalcoholic extracts was 4.3 ± 0.1.

### 2.6. Solid-Phase Extraction (SPE)

The hydroalcoholic extracts (**HA_com_**, **HA_sel_**) were subjected to solid-phase extraction using a Discovery^®^ DSC-18 SPE Tube column (Merck Life Science s.r.l., Milan, Italy) according to Cairone et al. [23] with some modifications. The column was previously activated with methanol and then, conditioned with water acidified with 5% of acetic acid. 1 g of hydroalcoholic extract was dissolved in 5 mL of water and loaded into the column. The column was washed with 5 mL of water acidified with 5% of acetic acid and then eluted with 5 mL of methanol and 5 mL of ethanol. The obtained fractions (**HA-SP_com_**, **HA-SP_sel_**) were concentrated under reduced pressure at 40 °C with a rotary evaporator, weighed and stored at 4 °C until analyzed.

### 2.7. Partitioning in Ethyl Acetate 

The hydroalcoholic extracts, (**HA_com_**, **HA_sel_**), dissolved in 10 mL of water, were extracted thrice in a separating funnel with 10 mL of ethyl acetate. The organic fraction (HA-EA_com_, HA-EA_sel_) was concentrated under reduced pressure at 40 °C with a rotary evaporator, weighed and stored at 4 °C until analyzed.

### 2.8. GC-MS Chemical Analysis

The chromatographic analyses of the peels and of all of the obtained extracts were carried out on Clarus 500 model Perkin Elmer (Waltham, MA, USA) gas chromatograph coupled with a mass spectrometer equipped with a flame ionization detector (FID) and a Varian Factor Four VF-1 capillary column. The operative conditions were the following: oven temperature at 40 °C for 2 min, then increased to 220 °C at 6 °C/min and finally held for 10 min at this same temperature (for the pulps and peels headspace); oven temperature program at 60 °C then increased to 170 °C at 6 °C/min, increased to 250 °C at 8 °C/min and finally held for 10 min at this same temperature (for the liquid phase of the extracts); injector temperatures: at 250 °C for the peels and 270 °C for the direct injection of the extracts liquid phase. 

Helium was used as carrier gas at a constant rate of 1 mL/min. The mass spectrometer was operated at 70 eV (EI) in scan mode in the range 40–400 *m*/*z*. Ion source and the connection parts temperature were 220 °C.

The identification of volatile compounds was performed by matching their mass spectra with those stored in the Wiley 2.2 and Nist 02 mass spectra libraries database and by calculating the Linear Retention Indices (LRIs) using a series of alkane standards analysed under the same conditions used for the samples. LRIs were then compared with available retention data reported in the literature. The peak areas of the FID signal were used to calculate the relative concentrations of the components expressed as percentage, without the use of an internal standard and any correction factor. All analyses were carried out in triplicate.

### 2.9. Antioxidant Activity by DPPH (2,2-Diphenyl-1-picryl-hydrazyl) Method

According to Cairone et al. [23] a solution 100 µM of DPPH was prepared in methanol. Then, 2 mL of this solution were added to 1 mL of methanol, stored in the darkness, and monitored by UV/VIS Lambda 25 spectrophotometer (Perkin Elmer Waltham, MA, USA), at the wavelength of 515 nm, until the absorbance value was stable. 

0.5 mL of a sample solution (5 mg/mL) were added with 2 mL of the same DPPH solution and 0.5 mL of methanol. The absorbance at 515 nm was controlled, following the same conditions described above, and the reduction of DPPH absorbance after 30 min was detected. Finally, a calibration curve was constructed to quantify the antioxidant activity by adding 1 mL of gallic acid (from 0.9 to 6.5 µg/mL), at different concentrations, to 2 mL of the DPPH solution, following the previous described conditions. A calibration curve was constructed (y = 0.6473e^−378.5x^) and the antioxidant capacity was expressed as gallic acid equivalents.

### 2.10. Antifungal Activity Assay

Growth inhibition assays were performed according to standardized methods for yeast using the broth microdilution method (CLSI M27-A3, 2008; CLSI, 2012) [24]. The strains *C. albicans* ATCC24433, coming from the American Type Culture Collection (ATCC, Rockville, MD, USA), and *C. glabrata* PMC0849, PMC0822, PMC0806 PMC0843, coming from the Pharmaceutical Microbiology Culture Collection (PMC, Sapienza, Rome, Italy), were tested. The strains were grown on Sabouraud dextrose agar (Sigma Aldrich, St. Louis, MI, USA) at 35 °C for 24 h. The suspension of *Candida* cells was prepared, and the final concentration of the inoculum was 1.0 × 10^3^–1.5 × 10^3^ CFU/mL (CLSI. M38-A2, 2008). The extracts were dissolved in dimethyl sulfoxide at concentrations 100 times higher than the highest tested concentration. The extracts were then serially diluted 2-fold across the 96-well plates in RPMI 1640 medium (Sigma-Aldrich, St. Louis, MI, USA) and the final concentration ranged from 1024 µg/mL to 0.5 µg/mL. The plates were incubated at 35 °C. After 24 h, the lowest concentration of extracts that caused ≥50% growth inhibition (MIC) was determined. The experiments were performed three times in duplicate. The results are expressed as median.

### 2.11. HPLC-DAD Analysis

Kiwi peels hydroalcoholic extracts were weighed, dissolved in methanol and analysed with a HPLC-DAD (Perkin Elmer, Milan, Italy), equipped with a Series 200 LC pump, a Series 200 DAD and a Series 200 autosampler, including a TotalChrom Perkin Elmer software for plotting data. The analyses were performed on a Luna RP-18, 3µ, with a linear gradient constituted by acetonitrile and water acidified by 5% formic acid, from 100% of aqueous phase to 35% in 55 min, at flow rate of 0.9 mL/min. Calibration curves were expressed in µg/mL and were constructed for catechin (y = 5.18 x–− 24.29; R^2^ 0.9997), epicatechin (y = 5.01x + 21.6; R^2^ 0.9995), caffeic acid (y = 26.03x + 20.37; R^2^ 0.9974), sinapic acid (y = 11.37x + 9.92; R^2^ 0.9984).

### 2.12. Statistical Analysis

Each assay was replicated at least three times. Data are expressed as mean ± SEM and statistical significance was determined using the XLStat 2021, software (New York, NY, USA).

## 3. Results and Discussion

Pomace, constituted by peels, seeds, and other parts resistant to the squeezing process, represents the primary by-product of the kiwi juice industry [25]. It represents a valuable source for the recovery of useful dietary fibers, as well as for other metabolites of health and economic values such as polyphenols, carotenoids, chlorophylls, and aroma compounds.

In the present work, a work-flow on the kiwi peels was studied with the aim to deep the knowledge about their aroma character, the potential added value deriving from the use as source of extracts and biomolecules for cosmetic and pharmaceutical applications. Finally, we aimed to evaluate the possible applications of the residue of the applied extraction, in view of the obtainment of zero impact by kiwi wastes.

### 3.1. SPME-GC/MS of Separated Kiwi Peels 

Comparison SPME-GC/MS analysis data of kiwi peels obtained from commercial and locally produced fruits (selected peels) shows a relatively simple aroma mixture in the latter (See also chromatograms, Appendix A), represented for more than 90% by linalool (50%), ocimenol (16%), *α*-terpineol (16%) and β-myrcene (10%). More complex results the aroma mixture from the commercial sample constituted by twelve molecules represent in percentages ranging between 24 and 3%, with α-terpineol being the most abundant (24%) followed by myrcenol (22%) and *p*-menth-1-en-9-al (8%). (See also Figure 1 and Appendix A).

Impact compounds on the kiwi aroma, as reported in the review by Garcia et al. [26], are usually obtained by simultaneous distillation extraction, vacuum distillation, and methods based on both SPME and head space analysis. Few molecule examples correspond to our experimental data if the previously reported results only refer to whole kiwi fruits. To our knowledge, no data are available on the direct application of this technique to separated kiwi peels.

Some of the identified molecules in the peels, such as linalool, *α*-terpineol, myrcenol, ocimenol, and *β*-myrcene, are significant volatile compounds and are especially correlated with wines and other grape derivatives [27] which are used in food and perfumery aroma industry. 

To our knowledge, no data are available in the literature concerning the obtainment of essential oils by kiwi peels and relative production yield, aspects which deserve to be investigated further. Regardless, kiwi peels’ pleasant flavor promotes their employment in different application fields (for example, as dietary fibers food supplements).

### 3.2. Bioactives Extraction and Waste Recycle by Separated Kiwi Peels

Several techniques have been reported for the extraction of bioactive compounds from kiwifruit, its by-products, and different parts of the plant. Environmentally friendly solvents such as water or ethanol are recommended for the extraction, especially if the product will be used for food and nutraceutical applications. In addition, these technologies are adequately efficient and allow short operating times [28].

In this work, the fresh kiwi peels coming from the local production or from the market samples were prepared and extracted by a double phase extraction, as reported in our previous paper [20]. The simultaneous presence of a carotenoid and chlorophyll significant fraction, in addition to the polyphenolic component, imposes the use of organic solvents other than an hydroalcoholic mixture. This one-pot approach allowed us to save time, money, and solvent quantities, thereby reducing as much as possible the used unsafe *n*-hexane (which was chosen as the best compromise respect to other organic and chlorinated solvents), giving excellent results in terms of yield and stability of obtained extracts.

The yields from hydroalcoholic extraction resulted in 7.5 and 7% of the fresh weight, respectively, for locally produced (**HA_sel_**) and commercial sample (**HA_com_**). Considering a water content of about 78% (known by dehydration experiments performed on the same kiwi peels), the yield in hydroalcoholic extract was about 33% for dry peels. According to Martín-Cabrejas et al. [25], but also considering the simultaneous presence of ethanol as extraction solvent, in this fraction we should separate the polyphenolic fraction, soluble sugars content, ashes, and soluble fibers. A green color residue, furtherly evaluated by CIEL*a*b and confirmed by spectrophotometric analyses, indicates a small residue of chlorophyll and carotenoid content with respect to the organic extract.

The *n*-hexane extracts (**HE_sel_** and **HE_com_**) gave about a 0.05% yield and was mainly composed by chlorophylls and carotenoids. In this case, considering a water content of about 78%, the yield in *n*-hexane extract was about 0.22% and the presence of chlorophylls and carotenoids was confirmed by CIEL*a*b, spectrophotometric and HPLC analysis.

In order to isolate the polyphenolic fraction from the hydroalcoholic extract, further extraction and purification steps were carried out, such as a solid-phase extraction on the two cultivars (samples **HA-SP_sel_** and **HA-SP_com_**) with both affording an extraction yield of 1%. 

A partitioning between water and ethyl acetate was also performed (samples **HA-EA_sel_** and **HA-EA_com_**), affording smaller yields of about 0.1%. 

Performing these two kinds of further extraction, we obtained watery fractions representing about the 99% (SPE technique) or the 99.9% (water/ethyl acetate repartition) of the initial hydroalcoholic extract. These fractions represent furtherly purified water-soluble fibers, respectively obtained as first eluted phase by SPE column or by partitioning.

Although our attention has been in the past mainly directed towards the polyphenolic fraction, which here represents the remaining 1 or 0.1%, the waste fraction which is rich in soluble fibers is not negligible. In fact, dietary fibers, although not absorbed, are considered a vital nutrient for the organism, as they have been shown to play a role in the maintenance of a good health state, favoring peristalsis and preventing of several illnesses such as colon inflammation and cancer, blood and cardiovascular diseases, diabetes, and other related comorbidities [29,30].

This waste fraction, rich in soluble fibers (containing soluble hemicelluloses, gums, mucilages, and pectin substances), could also be used in the formulation of supplements, due to its role in increasing viscosity, as well as the insoluble fibers (rich in lignin, cellulose, chitosan and insoluble hemicellulose). From a recycling perspective, it could be used as a substrate for the preparation of nanocellulose (NC). Cellulose, in the form of nanostructures, is considered one of the most promising green materials of recent acquisition [31]. NC materials have many advantageous properties, such as chemical inertness, excellent mechanical properties, large specific surface area, and the availability of numerous hydroxyl groups that can be readily functionalized via chemical reactions [32]. 

All of these properties make nanocellulose a promising material for a multitude of applications in the biomedical and engineering fields (and in many other emerging fields). In the biomedical field, owing to its biocompatibility, low cytotoxicity, and tunable surface features, it is becoming increasingly popular for the preparation of hydrogels, scaffolds for tissue engineering, and innovative drug delivery systems. Particularly, in the field of drug delivery it can be used as excipient (but also as starting matrix) to prepare NC-based oral, transdermal and local drug delivery systems, which are able to control drug release and improve both drug stability and therapeutic effects [33,34,35,36].

The obtained extracts were then subjected to the above reported GC-MS analysis (Figure 1), to HPLC-DAD and to DPPH assay for the evaluation of anti-radical capacity. The GC-MS analysis on the hydroalcoholic extracts, both showing a prevalence of Maillard products, reflects the greater complexity of the commercial respect to the locally produced kiwifruits, also confirmed by the furtherly purified extracts (HA-EA, HA-SP). A relatively simple mixture was shown by the *n*-hexane extracts (HE).

On the basis of the afforded yields and subsequent results by HPLC analysis and DPPH assay, a scheme of the better recycling is proposed in Figure 2.

### 3.3. Colorimetric Analysis

CIEL*a*b data, obtained from colorimetric analyses of kiwi hydroalcoholic and *n*-hexane extracts and the resulting reflectance profile curves are shown in Figure 3.

The lightness, L* values, contained in a narrow range, between 72 and 77, show samples characterized by a high brightness. The a* values, negative for sample greenness and ranging around −7 in the HA and −10 in the HE extracts, suggest the chlorophylls’ pigments. Nevertheless, the yellowish appearance of the hydroalcoholic extracts are represented in both types of samples. Spectrophotometric measures (not reported) confirmed the slight presence of chlorophylls pigments, highly concentrated in the kiwi samples, which were also retained in the hydroalcoholic extracts (in which they represented about one tenth respect to the *n*-hexane extracts). The curves related to the extracts in *n*-hexane show a trend at 660 nm, which is associated with a higher content of darker green pigments.

The very high b* positive values, ranging between 54 and 57 in the hydroalcoholic and between 83 and 92 in the *n*-hexane extract, could be related to a significant polyphenolic and carotenoid fraction, respectively. Carotenoid fraction was also quantified by the HPLC analyses, which showed the presence of about 28 mg/g of carotenoids dry extract. Carotenoids were quantified as lutein equivalents in hexane extracts. (See Appendix A).

No differences were evidenced from a colorimetric point of view between the locally produced and the commercial analyzed samples. It was not possible to compare the obtained results with those reported in the literature, since the only available papers analyzed the color of whole or sliced fruits, evaluating the browning after storage or thawing [37,38]. 

### 3.4. DPPH Assay

Many diseases such as Alzheimer’s disease, inflammation, atherosclerosis and Parkinson’s disease are associated to reactive oxygen species (ROS) production, which can damage the biological macromolecules, generating radical chain reactions [39].

In our studies, we used DPPH assays to evaluate the radical scavenging activity of kiwi peels hydroalcoholic extract (**HA_sel_**, **HA_com_**, **HA-SP_sel_**, **HA-SP_com_**, **HA-EA_sel_**, and **HA-EA_com_**). To perform the analyses, a DPPH solution was monitored in the darkness at room temperature until its absorbance values at 515 nm was stable. Afterwards, a known concentration of the different extracts was added and its antiradical activity was evaluated by spectrophotometric analysis. The obtained results, expressed as mg equivalents of gallic acid/g extract (mg GA/g), are reported in Figure 4. As shown therin, the samples coming by SPE purification presented the highest DPPH value (53.3 mg GA/mL in **HA-SP_com_** and 42.1 mg GA/g in **HA-SP_sel_**), followed by **HA-EA_com_** and **HA-EA_sel_** (26.7 mg GA/g and 37.7 mg GA/g respectively). The lowest DPPH values were recorded for hydroalcoholic extracts (7.2 mg GA/g in **HA_com_** and 12.1 mg GA/g in **HA_sel_**). This is due to the lower analyte concentration in the hydroalcoholic extract, which justifies the lower antiradical activity (as further confirmed by HPLC analysis). The HA-EA fractions result in lower antiradical activity and lower extraction yield compared to the extracts obtained by SPE, suggesting a limited efficiency of the ethyl-acetate partitioning process for the massive recovery of bioactive purified fraction. Our present findings suggest selecting only the SPE as a key role in the proposed scheme of pomaces recycling, giving higher yields and higher anti-radical activities with less of an impact on time and solvents.

Several DPPH assays, related to kiwi peels and available in the literature, reported values in agreement with our results by SPE purification, highlighting a strong antioxidant power and confirming the high added value of this waste materials [4,39,40,41,42].

Our results overlap with those from Fiorentino et al. [4], which reported a% inhibition of DPPH ranging between 60 and 90% for ethanolic extracts by Haywards kiwi peels. Our results, analogously calculated, ranged between 70 and 80% in HA-SP and HA-EA extracts. Hădărugă et al. [42] reported different results (22–80% inhibition) according to the hydroalcoholic composition, with values rising with ethanol percentages. These results are also in agreement with the inhibition shown by ours HA extracts, ranging from around 20 to 25%.

### 3.5. Anti-Candida Activity

The antifungal activity of kiwi extracts was tested in vitro against one strain of *Candida albicans* and four strains of *C. glabrata.* An interesting activity, growing from the raw hydroalcoholic extracts towards the more purified SPE and ethyl acetate extracts, was revealed. Moreover, the extracts obtained from local cultivation were more effective.

The strongest activities, characterized by MIC_50_ ranging between 4 and 32 µg/mL, were shown by the ethyl acetate extracts coming from this selected cultivar. Of particular interest was the efficacy, at 4 µg/mL, against PMC806 and PMC843 strains, which are more resistant to fluconazole, (MIC_50_ of 2 µg/mL). Promising activity was also exerted on C. albicans (MIC_50_ 8 µg/mL vs. 2 µg/mL of fluconazole),

These results (Table 1), on the whole, seem to be relevant for potential application of kiwi ethyl acetate extracts in antifungal therapies, always remembering to pay attention to the phytocomplex quali-quantitative composition.

### 3.6. HPLC-DAD Analysis

Crude and purified extracts obtained from the kiwi peels were subjected to HPLC-DAD analysis, and a chromatogram was recorded at 280 nm for the identification of hydroxycinnamic acids (caffeic acid and sinapic were found and quantified) and flavanols, (catechin and epicatechin were found and quantified) and at 360 nm for the identification of flavonols. However, no peaks were shown at this last wavelength (See also Appendix A).

As shown in Figure 5 and Table 2, the catechin amount varied between 3 and 24 mg/g of the extract (minimum value detected in **HA_com_** and maximum in **HA-SP_sel_**). Epicatechin, between 3 and 9 mg/g, was only recorded in SPE purified extracts and was more represented in the commercial sample. Caffeic acid varied between 0.5 and 9 mg/g, and along with sinapic acid (1.6–1.7 mg/g) was found only in SPE extracts.

The obtained results, only partially agree with those reported in the literature. Satpal et al. [2], in a review, confirms the main presence in peels from various kiwi cultivars, of caffeic acid and its derivatives, sinapic acid, catechins and derivatives. Other works also report a small presence of flavonoids, such as chrysin and quercetin [16,43], not yet identified in our extracts.

Very variable values are reported in literature according to the analyzed cultivars, mostly expressed as mg/g fruit fresh weight. The sum of reported flavanols contents ranged between 0.003 and 0.148 mg/g fw and hydroxycinnamic acids between 0.01 and 0.7 mg/g fw. Taking in to account the differences due to the expression in fresh weight respect to dry weight, the available quantitative data, only partially overlapped with our results [16,44]. The presence of quercetin is sometimes reported, but was not found in our samples.

Comparing data, a strong analytes concentration was afforded in SP extracts, both of commercial and selected kiwi fruits, corresponding to about 19-fold increase in catechin and 6-fold increase in caffeic acid. Moreover, this concentration results in the presence of epicatechin and sinapic acid extracts, not identified in hydroalcoholic extracts. The presence of a more active phytocomplex in SPE extracts, justifies the higher anti-radical capacity shown by DPPH analyses.

As regards the difference between the phytocomplexes of the two samples, the main difference is represented by epicatechin that is present in concentration three folds higher in the peels from commercial sample, perhaps in accordance with the stronger anti-radical activity expressed.

## 4. Conclusions

In this study, a green and complete approach for the recycling of kiwi peels was suggested, affording the potential recovery of tons of wasted material to be reused as high added value new compounds.

Considering that millions of tons of kiwi waste worldwide are thought to be produced in a year, the proposed extraction method could afford amounts in the order of tons per year of extracts enriched in carotenoid, chlorophylls, and polyphenolic extracts.

Insoluble and soluble fibers, representing the main part of wastes, could also be recycled and used as interesting emerging materials for pharmaceutical and other health uses. Moreover, considering the real content of analytes and taking in to account the principles of the circular economy and the best green procedures, the hydroalcoholic extracts could also represent an interesting compromise. In the detailed case we report, it contains about 100 mg/100 g catechin and 16 mg/100 g caffeic acid, still retaining a pleasant aroma by HMF and derivatives. It could be easily formulated as an enhanced mix of soluble fibers and antioxidant compounds or as a food supplement.

Moreover, the ethyl acetate extracts, although affording estimated quantities of 220 tons per year, deserve to be taken in to account as potential anti-*Candida* agent.

## Figures and Tables

**Figure 1 foods-11-00589-f001:**
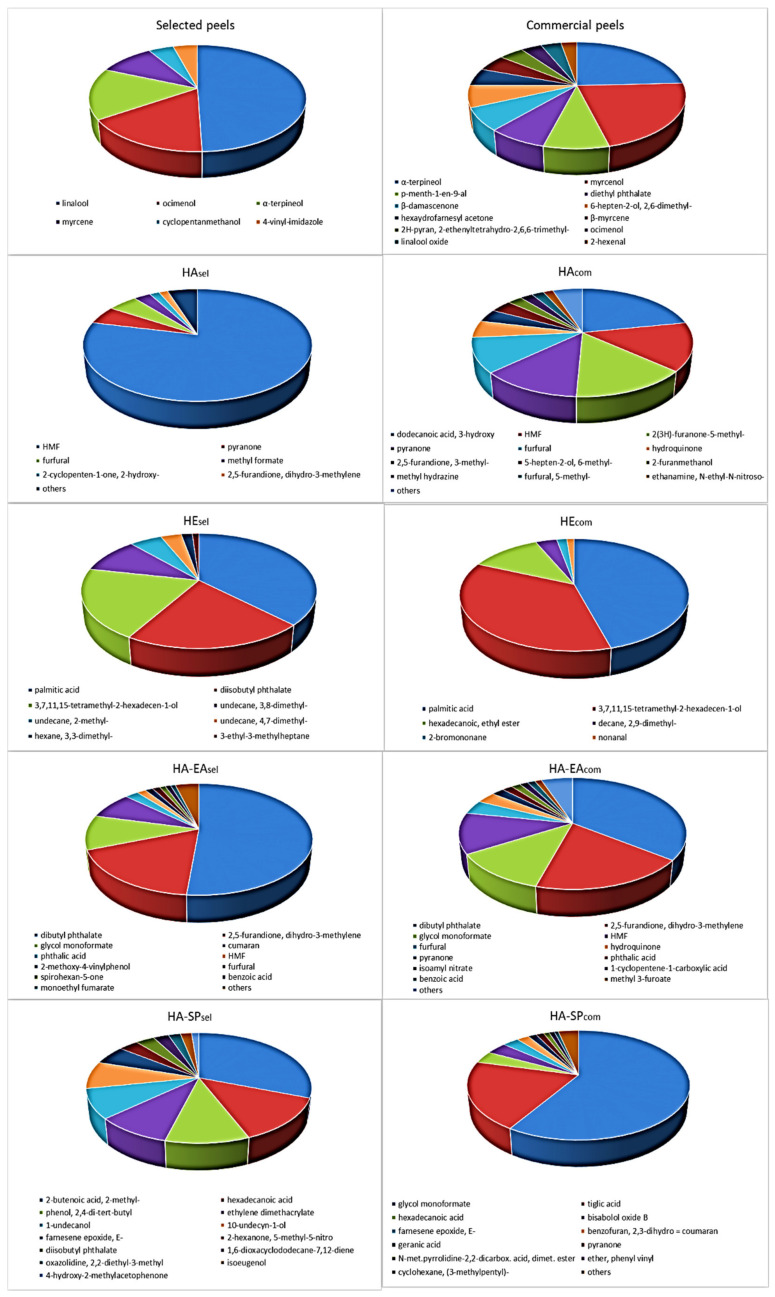
GC-MS and SPME-GC-MS analyses of the most significant components of kiwi peels and extracts.

**Figure 2 foods-11-00589-f002:**
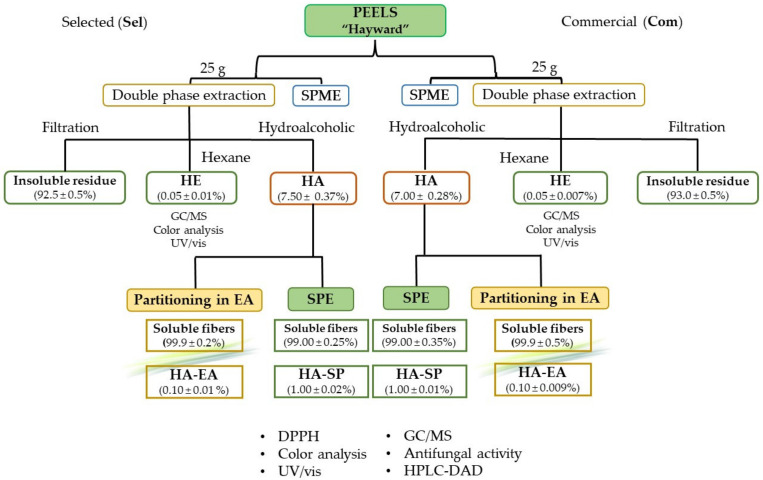
Scheme of peels recycling and indicative estimate of the obtained quantities.

**Figure 3 foods-11-00589-f003:**
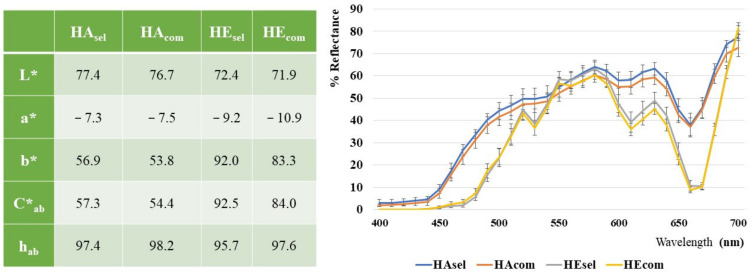
Colorimetric data and reflectance curves of the hydroalcoholic (HA) and organic (HE) extracts. The RSD values, evaluated on triplicates, were <5%.

**Figure 4 foods-11-00589-f004:**
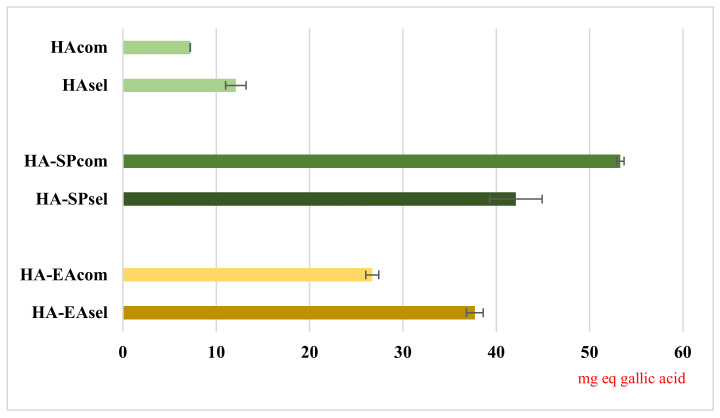
Comparison of anti-radical capacity related to different extraction methods. Values are expressed as mg equivalents of gallic acid/g of obtained extract.

**Figure 5 foods-11-00589-f005:**
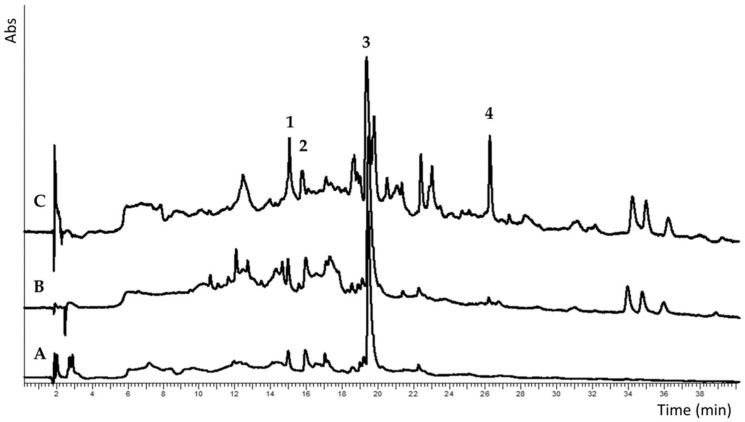
Example chromatograms at 280 nm of: A, hydroalcoholic extracts; B, HA-SP extracts; C, HA-EA extracts. Quantified peaks: 1, catechin; 2, epicatechin; 3, caffeic acid; 4, sinapic acid.

**Table 1 foods-11-00589-t001:** Antifungal activity of kiwi samples. Activity was determined according to Clinical and Laboratory Standards Institute guidelines (CLSI document M38-A2, 2008). Minimal inhibitory concentration (MIC) was determined. MIC_50_, the lowest drug concentration that prevented 50% of growth with respect to the untreated control. The values shown are the median from three independent measurements).

	*C. glabrata*				*C. albicans*
	Median MIC µg/mL				
	PMC0849	PMC0822	PMC806	PMC843	ATCC24433
**HA_com_**	512	256	512	512	256
**HA_sel_**	512	128	256	192	256
**HA-SP_com_**	512	128	256	128	512
**HA-SP_sel_**	256	128	128	128	256
**HA-EA_com_**	64	32	32	32	32
**HA-EA_sel_**	32	4	4	4	8
**Positive control (Fluconazole)**	0.5	0.5	2	2	2

**Table 2 foods-11-00589-t002:** HPLC-DAD data of the obtained peel kiwi extracts. The results are expressed in mg/g of dry extract. BLD, below limit of detection.

	Catechin	Epicatechin	Caffeic Acid	Sinapic Acid
**HA_com_**	3.2 ± 0.4	BLD	0.5 ± 0.01	BLD
**HA_sel_**	2.9 ± 0.1	BLD	0.5 ± 0.02	BLD
**HA-SP_com_**	21.8 ± 3.1	9.4 ± 1.2	6.1 ± 0.8	1.7 ± 0.3
**HA-SP_sel_**	23.6 ± 4.3	2.8 ± 0.9	6.1 ± 0.6	1.6 ± 0.1
**HA-EA_com_**	10.1 ± 2.4	BLD	5.9 ± 0.9	-
**HA-EA_sel_**	16.4 ± 1.1	BLD	8.3 ± 0.1	-

## Data Availability

Not applicable.

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
