# Peer review of "Valorization of Kiwi Peels: Fractionation, Bioactives Analyses and Hypotheses on Complete Peels Recycle"

_foods, 2022, doi:10.3390/foods11040589_

Round 1

Reviewer 1 Report

Work focused on the fractionation of kiwi fruit peels. Color, antioxidant and antimicrobial activity are analyzed, and a comparison is made between samples obtained from the market and those obtained directly from a plantation. There are many aspects that should be improved before considering publishing this work in this journal. 
- I do not believe that the title represents what is studied in the paper. There is no selective extraction. It is a fractionation. The assumptions of the use of pomace is circumstantial. Only the skin is analyzed in this paper and to transfer the result to an industrial scale is too bold. 
- Although I consider that I am not qualified to judge English language and style, the first sentence of the paper include 5 lines (29-33). This is not usual in the English language. In addition there are sentences that are difficult to understand. I suggest a revision of the language. 
- The introduction should be revised. There are repetitive aspects. The most commercialized fruit... is discussed in at least three paragraphs. Several varieties are discussed (A. chinensis and A arguta) that are not studied in this work. These three are mentioned as having antioxidant properties... The other varieties do not? Paragraphs 86-106 should be reduced. There are times when previous works are discussed but it is not known if references are missing or are those that are several lines further back (e.g. lines 107-111 and references 15-18). Italy is the first producer is commented in two parts of the text. 
- Section 2.2. Data should be given on the moisture content of the samples and whether they are reduced in size. What is the final average size of the samples? In this section two types of samples are taken. It is convenient to indicate from here that the subscript SEL or COM will be used for each. 
- When an acronym is cited for the first time, its meaning should be indicated. For example SPME.
- It is not clear what colorimetric analysis is for. The conclusions obtained from this analysis are not supported by this type of analysis. Obviously, if kiwifruit is extracted, and the sample is green, we will have chlorophyll.... It is strongly recommended to use standard methods for the quantification of chlorophylls, carotenoids, total polyphenols... 
- Section title 2.9. Better "Antioxidant activity by DPPH method".
- I suggest putting a block diagram of the process that has been performed on each sample, with the overall yields of each step for each case. 
- Section 3. Best "Results and discussion".
- Section 3 begins by discussing the applications of pomace, but only a part of this is discussed in the paper. In addition, pomace is produced only juice generation. Domestic consumption only generates peel. Some conclusions of the paper are made on pomace and others only from the peels. This should be avoided in the paper. If only the skin is analyzed, the pomace conclusions should be avoided. 
- I don't know if lines 274 and 284 say different things or the same thing. 
- In section 3.1 no quantitative values are given, only %. It is difficult to conclude that it is a raw material to produce essential oils (line 276). 
- It is difficult to analyze Figure 1. One objective is to compare COM with SEL. Maybe it is good to include several chromatograms to see the dispersion in COM regarding to SEL. 
- About the analysis in figure 1. The first graph I suppose is the complete material... the remaining ones are the fractions. If in the first graph we get some compounds, should they appear in the various fractions...? Without a block diagram of what has been done I can't understand the work. Perhaps these data would be better in tables, with their error intervals and statistical analysis.
- Section 3.2. The authors in the text present numerous data, without error intervals. All these data should be included in a table or graph and in the discussion refer to it. 
- In this section it is indicated that one of the fractions is rich in fiber. An important part of the text is dedicated to discussing the potential of fiber. But the amount of fiber in the sample is not measured in the manuscript. 
- I do not understand Figure 2. Are the data presented supported by the data in the paper?. It talks about pomace, when only the skin has been studied. 
- Line 209-411. It is commented that the data in the bibliography agree with those obtained. Three references are cited. The analysis should have been extended a little more. Reference 41 is in juice, reference 40 is in skins, and it seems that they obtain higher results. 
- In Figure 4 it should be indicated in the figure what is represented on the X-axis with units.
- In figure 5. Four compounds are identified, but there are numerous peaks that are not identified. 
- Table 2 and Figure 6 represent the same. I suggest only table 2.

Reviewer 2 Report

This study reveals the potential uses of peel kiwi extracts for food and nutraceutical applications.

The work seems to be well planned and carefully executed, and the data were well explained.

Minor comments:

Line 162: define the term HAcom and HAsel (first time that appears in the text);

Line 189 and 191: introduce the unit ºC in 6º/min;

Line 217: explain the use of an exponential calibration curve between absorbance and concentration.

Line 403: correct the term HE-EA

Table 2: define BLD-

Reviewer 3 Report

Utilization of fruit waste and by-products has a high relevance for the industry and correspond to the main principles of circular economy. Kiwi peel is a good source for the extraction of bioactive components. Development of appropriate extraction methods, investigation of their efficiencies, and analysis of extracted components can provide interesting information for the readers. Therefore, the topic of the manuscript can be considered as relevant and interesting.

General opinion:

The manuscript is generally well structured. Introduction section summarizes well the relevancies and the novelties of the study (complete recycling conception) and research motivations. Applied analytical methods are adequate to the sample characteristics. Materials and methods are described clearly. The manuscript contains interesting and valuable results that are represented clear in figures and tables and discussed with relevant references.

Specific comments, questions:

Line 14-16: Please give why the kiwifruit peel suitable a good source for polyphenol extraction (instead of giving the source os samples).

Please give the details of colorimetric analysis (temperature, pH, etc).

The visibility of Figure 1 is very poor. Please give the aroma component concentration and yield, as well.

How can generalize the quantities related to pomace recycling (if authors investigated just one convarietas)?

Please improve the quality of Figure 3 (and check the typos, ’wavelenght’, for instance).

Please check the axis title position in figures (see Fig. 3 and 6, for instance).

Round 2

Reviewer 1 Report

The manuscript has been improved enormously. Figure 2 gives very useful information about the work done. Regarding to Figure 1, I still don't like it, but I can't think of any other way to provide these information. The supplementary material included overcomes this problem. The original manuscript concluded some issues that were not supported by the data. In this version they have been eliminated. Conclusions regarding compositions in the colorimetric analysis have also been avoided. My proposal is to accept the paper in present form.

Reviewer 3 Report

The manuscript has an interesting and relevant topic. Authors have revised the MS thoroughly according to reviewers' comments and suggestions. Amendments, rephrasing, additional data, change of the title, clear discussion of experimental data made the amnuscript more complete and clear. The overall scientific quality of the manuscript has been improved significantly due to the revision.

Author Response

We would like to thank the reviewer for his kind reply.